# Infiltration of Conduction Tissue Is a Major Cause of Electrical Instability in Cardiac Amyloidosis

**DOI:** 10.3390/jcm12051798

**Published:** 2023-02-23

**Authors:** Andrea Frustaci, Romina Verardo, Matteo Antonio Russo, Marina Caldarulo, Maria Alfarano, Nicola Galea, Fabio Miraldi, Cristina Chimenti

**Affiliations:** 1Cellular and Molecular Cardiology Lab, IRCCS L. Spallanzani, Via Portuense 292, 00149 Rome, Italy; 2MEBIC Consortium, San Raffaele Open University, 00166 Rome, Italy; 3Cellular and Molecular Pathology, IRCCS San Raffaele, 00166 Rome, Italy; 4Department of Clinical Internal Medicine, Anesthesiology and Cardiovascular Sciences, Sapienza University, 00185 Rome, Italy; 5Department of Experimental Medicine, Sapienza University, 00185 Rome, Italy

**Keywords:** conduction tissue, cardiac amyloid, arrhythmias

## Abstract

Background: The pathology of conduction tissue (CT) and relative arrhythmias in living subjects with cardiac amyloid have never been reported. Aims: To report CT pathology and its arrhythmic correlations in human cardiac amyloidosis. Methods and Results: In 17 out of 45 cardiac amyloid patients, a left ventricular endomyocardial biopsy included conduction tissue sections. It was identified by Aschoff–Monckeberg histologic criteria and positive immunostaining for HCN4. The degree of conduction tissue infiltration was defined as mild when ≤30%, moderate when 30–70% and severe when >70% cell area was replaced. Conduction tissue infiltration was correlated with ventricular arrhythmias, maximal wall thickness and type of amyloid protein. Mild involvement was observed in five cases, moderate in three and severe in nine. Involvement was associated with a parallel infiltration of conduction tissue artery. Conduction infiltration correlated with the severity of arrhythmias (Spearman rho = 0.8, *p* < 0.001). In particular, major ventricular tachyarrhythmias requiring pharmacologic treatment or ICD implantation occurred in seven patients with severe, one patient with moderate and none with mild conduction tissue infiltration. Pacemaker implantation was required in three patients, with complete conduction section replacement. No significant correlation was observed between the degree of conduction infiltration and age, cardiac wall thickness or type of amyloid protein. Conclusions: Amyloid-associated cardiac arrhythmias correlate with the extent of conduction tissue infiltration. Its involvement is independent from type and severity of amyloidosis, suggesting a variable affinity of amyloid protein to conduction tissue.

## 1. Introduction

Cardiac amyloidosis (CA) is an infiltrative heart muscle disease characterized by interstitial myocardial deposition and polymerization into 10 nm-wide fibrils of amyloidogenic proteins, including antibodies’ light chains, serum amyloid A, transthyretin and apolipoprotein A1–A2 [1,2,3]. Βeta-pleated organization confers to amyloid fibrils specific physic-chemical properties such as uptake of Congo Red stain and birefringence to polarized light that are used for its histologic recognition. The accumulation of amyloid fibrils in the myocardium results in progressive thickening and stiffness of the cardiac wall with increasing compromise of diastolic and, finally, of systolic function. The decline in QRS voltages is commonly seen as the consequence of low amyloid conductivity, as well as the result of progressive cardiomyocyte loss. The involvement of cardiac conduction tissue (CT) is poorly investigated in CA because of the difficulty to obtain in-life histological sections of CT, although the occurrence of severe brady-arrhythmias requiring pacemaker implantation and/or ventricular tachy-arrhythmias resulting in sudden death is frequently registered [4,5,6,7]. We experienced the possibility, particularly in patients with thickened hearts, of including CT sections in endomyocardial biopsies obtained from the left ventricular septum [7,8,9,10]. In the following study, the pathology of CT affected by various forms of CA is reported.

## 2. Materials and Methods

### 2.1. Patient Population

From January 2011 to December 2021, 45 patients with a clinical phenotype of restrictive cardiomyopathy, according to the ESC definition [11] (characterized by myocardial hypertrophy leading to increased wall stiffness with elevated filling pressures and reduction in ventricular volumes), had a histological diagnosis of cardiac amyloid at left ventricular endomyocardial biopsy. Seventeen of them (eight male and nine females, mean age, 64.2 ± 8.9) had histological sections of CT included in at least 1 specimen. These 17 patients, presenting a variable type of amyloid protein, constituted our study population. The remaining 28 individuals with cardiac amyloid were excluded from the present study, as their biopsies failed to contain sections of CT. A study flow chart is presented in Figure 1. The study complies with the Declaration of Helsinki; the locally appointed ethics committee approved the research protocol and informed consent was obtained from all subjects.

### 2.2. Cardiac Studies

Extensive clinical examination, non-invasive (resting ECG, Holter monitoring, echocardiography with tissue Doppler analysis, cardiac magnetic resonance, nuclear scintigraphy with hydroxy bisphosphonate bone tracer 99mTc-HPD) and invasive cardiac studies were performed on all patients. Invasive cardiac examinations included cardiac catheterization, selective coronary angiography, left ventricular (LV) angiography and LV endomyocardial biopsy. Our center is a tertiary referring center for the diagnosis and treatment of cardiomyopathies, myocarditis and heart failure. Endomyocardial biopsy represents a common investigational approach for the histological and molecular characterization of myocardial tissue [8,12].

Endomyocardial biopsies (4 to 8 fragments) were performed in the septal-apical region of the LV. Myocardial samples were processed for routine histological and immunohistochemical analysis and for transmission electron microscopy. Endomyocardial biopsies were performed to obtain the definite diagnosis of cardiac amyloidosis in patients with clinical suspicion of AL amyloidosis or in patients with 99mTc-HPD bone scintigraphy Perugini score 1 or in the presence of Perugini score 2–3 and serum monoclonal component (increased in serum free-light chain, positive serum and/or urine immunofixation) [13].

We retrospectively analyzed the data of patients collected at baseline during recovery at our center; unfortunately, we cannot provide follow-up results.

### 2.3. CMR Image Acquisition and Analysis

CMR studies were conducted in 12 of the 17 patients (70%) by using a 1.5-T scanner (Magnetom Avanto, Siemens Healthcare, Erlangen, Germany) equipped with a multichannel phase-array cardiac coil. A standardized CMR protocol, including the ECG-gated cine steady-state free precession (cineMR) and late gadolinium enhanced (LGE) sequences, was performed as described [10].

Modified Look–Locker inversion recovery sequences (MOLLI) were acquired before and fifteen minutes after contrast injection in 8 out of 17 patients, because they were not available before 2015 in our MR scanner [10]. Myocardial extracellular volume fraction (ECV) maps were generated using the delayed post-contrast bolus, acquiring native and contrast-enhanced MOLLI images with identical slice location, matrix and spatial resolution. Image analysis was performed using a dedicated software (Cvi42, v5.3.0; Circle Cardiovascular Imaging Inc., Calgary, AB, Canada).

### 2.4. Histology and Electron Microscopy

For histological analysis, the endomyocardial samples were fixed in 10% buffered formalin and paraffin-embedded. Sections of 5 μm thickness were stained with hematoxylin and eosin, Masson trichrome and Congo Red stain. For transmission electron microscopy, additional samples were fixed in 2% glutaraldehyde in a 0.1 mol/L phosphate buffer, at pH 7.3, postfixed in osmium tetroxide and processed after a standard schedule for embedding in Epon resin. Ultrathin sections were stained with uranyl acetate and lead hydroxide. A Philips CM-10 transmission electron microscope was used for observation and photographic analysis. Immunostaining for the characterization type of amyloid proteins was performed on formalin/paraformaldehyde-fixed paraffin–embedded sections of endomyocardial samples using Human Light Chain Kappa and Lambda (AL type Amyloidosis), Human Serum Amyloid A (SAA), Transthyretin (TTR) and apolipoprotein A1-2 (Santa Cruz Biotechnology Inc., Dallas, TX, USA).

CT was identified at histology as loosely arranged small myocytes, positive to HCN4 immunostaining [7,9], supplied by a centrally placed thickened wall arteriole, circumscribed by a fibrous membrane in a fascicle configuration (Monckeberg and Aschoff criteria).

HCN4 immunohistochemistry was performed on formalin/paraformaldehyde-fixed paraffin–embedded sections of endomyocardial samples to identify HCN4-positive cells, as described [9]. Briefly, tissue sections were incubated with a rat monoclonal antibody at a dilution of 1:20 recognizing HCN4 (Pierce Antibody Products, Thermo Fisher Scientific Inc., Waltham, MA, USA) or without a primary antibody (negative control) overnight at 4 °C in a humidified chamber. Detection was performed using a biotin-conjugated secondary antibody and SA-HRP (UCs Diagnostic S.r.l., Morlupo, Italy) at room temperature for 10 min. Samples were washed and incubated by colorimetric detection using 3,3′-diaminobenzidine (Agilent Technologies, Inc., Santa Clara, CA, USA), counterstained with hematoxylin and mounted with a quick-hardening mounting medium (Eukitt, Bio-optica, Milano, Italy).

A quantitative evaluation of the severity of CT infiltration was assessed by counting the percent of CT area replaced by fibrous tissue and Congo Red+ material in at least 3 serial sections. The infiltration was defined as mild when ≤30% of the CT area was replaced by Congo Red+ material and fibrous tissue, moderate in >30% and ≤70% CT area involvement and severe in >70% CT area replacement. The extent of CT infiltration was correlated with age, MWT (maximal wall thickness), severity of arrhythmic manifestations (Lown class) and type of amyloid protein.

### 2.5. Statistics

Data were presented as mean ± standard deviation. Significance was set as *p* < 0.05. The correlation between CT infiltration and age, Lown class, MWT and type of amyloid protein was explored with Spearman’s rho test. Data analysis was performed using GraphPad Prism version 6.04 for Windows (GraphPad Software, La Jolla, CA, USA).

## 3. Results

Clinical data, amyloid characterization and the extent of CT infiltration in the 17 patients are summarized in Table 1. In particular, ECG was characterized by low QRS voltages and was associated to brady and/or tachyarrhythmias (panel A in Figure 2, Figure 3 and Figure 4). At echocardiography, LVMWT was increased in all patients (17.0 ± 2.4 mm), and different degrees of diastolic dysfunction were always present, while LV ejection fraction (EF) was preserved in all (51.4 ± 9.5%) but one patient, who developed an EF of 25% (Table 1). This patient showed at histology a remarkable infiltration by amyloid of intramural coronary vessels with severe lumen narrowing, suggesting overlapping ischemic damage.

Among 17 patients, 12 performed CMR and in one case contrast agent was not administered for chronic renal failure. CMR confirmed the presence of cardiac hypertrophy (panel B in Figure 2, Figure 3 and Figure 4) with preserved systolic function in all patients but one.

LGE was present in all patients, predominantly with diffuse (5/11) or subendocardial (4/11) patterns, compared to focal (2/11) (panel C in Figure 2, Figure 3 and Figure 4). In 7/11 patients, T1 mapping sequences were acquired; as expected, nT1 and ECV were increased in all patients (nT1: 1171 ± 61 ms, normal range in our center 990 ± 35 ms; ECV: 59.9 ± 7.5%). At cardiac catheterization, LV end-diastolic pressure was usually elevated (>12 mmHg). Coronary angiography showed normal coronary arteries in all subjects.

LV endomyocardial biopsy was a safe procedure. We did not record major complications or arrhythmias related to the procedure.

The diagnosis of cardiac amyloid was always sustained by LV endomyocardial biopsy showing the interstitial deposition of apple-green birefringent material at the polarized light of Congo Red-stained myocardial sections (insert in panel B of Figure 2, Figure 3 and Figure 4) and the ultrastructural recognition of characteristic 100 Ȧ thick fibrils (insert in panel C of Figure 4). At immunohistochemistry, amyloid was of AL type in thirteen patients and of TTR type in four (Table 1). CT was identified in all patients as histological sections that met the Monckeberg and Aschoff histological criteria and was positive to HCN4 immunostaining (panels D–F in Figure 2, Figure 3 and Figure 4). Of seventeen patients included in the study, five had mild, three moderate and nine severe CT involvement. Overall replacement of the CT area was 59 ± 28% (Table 1). CT was replaced by fibrous tissue and Congo Red-positive material. The extent of CT infiltration was paralleled by an analogous involvement of the CT artery. In the cohort of severe CT infiltration, the CT artery appeared remarkably narrowed (arrows in panels D and E of Figure 4), suggesting a possible ischemic contribution to the CT damage. The extent of CT involvement correlated with the severity of arrhythmic manifestations (Lown class) (Spearman rho = 0.8, *p* < 0.001), but not with age, LVMWT or type of amyloid protein. Specifically, patients with mild CT involvement (<30% CT cell replaced) manifested no major arrhythmias (Lown class 1) requiring pharmacological control and/or PM/ICD implantation, even in the presence of remarkable myocardial thickening (LVMWT ≥ 16 mm). In the cohort with moderate CT involvement (>30% and ≤70% cell replacement), an anti-arrhythmic therapy was needed in one patient with non-sustained ventricular tachycardia (panel A, Figure 3), and one patient received an ICD because of severe left ventricular dysfunction. In the nine subjects with severe CT infiltration, three required PM implantation and one an ICD, and five received anti-arrhythmic therapy. A specific chemotherapy including steroids and melphalan was adopted in those patients with AL amyloid and a monoclonal gammopathy or a multiple myeloma. No CT section was detectable at electron microscopy.

## 4. Discussion

CT involvement in CA is expected to be common on the base of frequency of brady and tachyarrhythmias occurring in the CA population requiring anti-arrhythmic treatment and/or PM/ICD implantation [13,14]. However, its documentation is usually provided postmortem, as investigation into the life of CT is considered unobtainable [15].

In this regard, we reported in previous studies [9,10] the possibility to include sections of CT by drawing multiple (five to eight) endomyocardial samples from the middle-lower portion of the left ventricular septum. This observation followed an extensive investigation into the left ventricular endomyocardial biopsy [8], suggesting it is as safe as the right ventricular one.

In the present report, CT was included in the biopsy samples of 17 out of 45 (37%) consecutive patients with CA. CT was histologically identified by the morphological criteria introduced by Aschoff and Monckeberg in 1910 and consisted in loosely arranged small myocytes supplied by a circle artery, surrounded by a fibrous membrane in a fascicle configuration by the specific molecular expression of HCN4. CT infiltration by CA is theoretically believed to faithfully reflect the involvement of the whole heart. Indeed, histological findings reported in our study suggest that CT can be affected independently from the degree of the left ventricular maximal wall thickness (LVMWT), with cases of mild to moderate CA (LVMWT ≤ 15 mm) replacing >70% of CT cells and severe forms (LVMWT ≥ 16 mm) involving < 30% of CT cells. The degree of CT involvement reflected in the absence of arrhythmias in the group with mild disease (CT cells involved ≤ 30%), low incidence of rhythm disturbances in cases of moderate CT involvement (>30% and <70% cell replacement) and remarkable incidence of conduction delay/block or supraventricular and/or ventricular arrhythmias in the cohort with severe CT involvement (>70% cell infiltration) requiring the institution of antiarrhythmic therapy and/or PM/ICD implantation.

Notably, CT involvement was paralleled by the analogous infiltration of CT-supplying coronary arteries. In particular, in the cohort with severe disease, CT vessels were massively infiltrated (Figure 4), suggesting the contribution of ischemic damage to CT deterioration. To summarize, CT and the rest of the myocardium seem to be separate compartments that may differently allocate CA deposition. This is because CT is bordered by a fibrous membrane and characterized by specific molecular structures such as HCN4, to which amyloidogenic proteins may manifest a different affinity. In this regard, amyloid preference toward a specific tissue is believed to follow chemical bonds, such as hydrogen ones, that favor one organ to others.

Likewise, no correlation was observed between the degree of CT infiltration and the type of amyloidogenic protein. In particular, no major differences in terms of the severity of CT infiltration were noted among AL and TTR amyloid deposition. It seems that single-mutated proteins might present specific a physic-chemical affinity for the molecular components of CT, promoting an elective amyloid deposition.

In conclusion, our study demonstrates that the occurrence of major brady or tachy-arrhythmias in patients with cardiac amyloidosis reflects a severe CT infiltration and requires an appropriate investigation, even with an electrophysiological study, in order to provide a prompt correction with anti-arrhythmic drugs and/or PM/ICD implantation. In addition, new drugs such as Tafamidis [16,17], Patisiran [18], Inotersen [19] and Diflunisal [20] have been recently introduced for the treatment of TTR amyloid, which may change the outcome of this disabling and deadly disease.

## 5. Limitations of the Study

The long enrolment time with different therapeutic options between patients enrolled at the beginning of the study compared to those enrolled recently provides a heterogeneous cohort that makes difficult the interpretation of response to therapy. Molecular mechanisms that favor an elective deposition of amyloid into CT are not yet clarified. The appearance of electrical instability in patients with CA should be taken into strong consideration for PM or ICD implantation, as it is the harbinger of grim prognosis.

## 6. Conclusions

CA-associated arrhythmias correlate with the severity of CT infiltration. CT involvement seems to be independent from type and severity of CA, suggesting a variable affinity of amyloidogenic protein to CT. Life-threatening cardiac arrhythmias may occur during all the progressive phases of human CA.

## Figures and Tables

**Figure 1 jcm-12-01798-f001:**
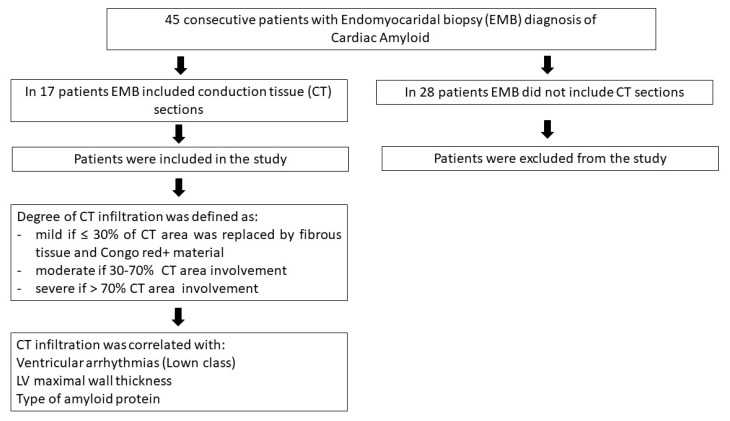
Flow chart of the study.

**Figure 2 jcm-12-01798-f002:**
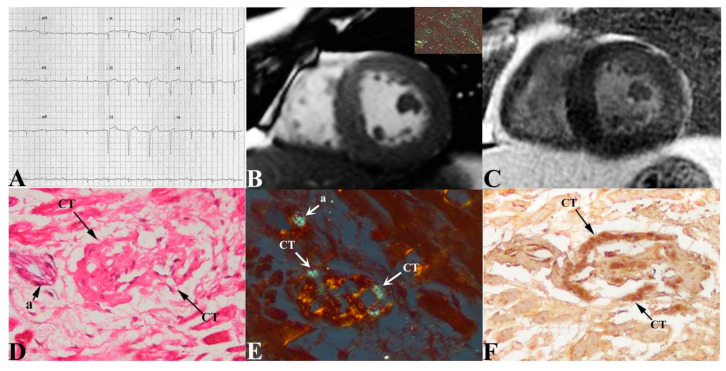
Mild infiltration of CT in cardiac amyloid. Low QRS voltages with absence of brady and/or tachyarrhythmias (**panel** (**A**)) are associated at CMR with severe LV hypertrophy (**panel** (**B**), MWT 20 mm). CineMR images show some slight “patchy” areas of LGE at mid-ventricular lateral wall (**panel** (**C**)). Apple-green birefringent material at polarized light of Congo Red-stained myocardial sections is visible in insert in panel B. Histology (**panel** (**D**), hematoxylin and eosin 400×), Congo Red staining (**panel** (**E**) 400×) and immunohistochemistry for HCN4 (**panel** (**F**), 400×) show mild infiltration of CT (conduction tissue) (arrows). a = arteriole CT = conduction tissue.

**Figure 3 jcm-12-01798-f003:**
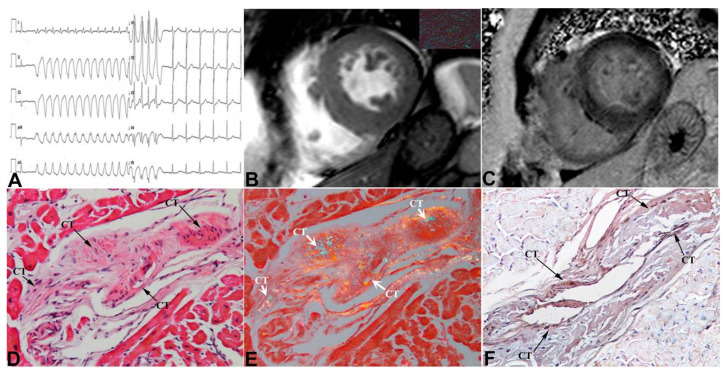
Moderate infiltration of CT in cardiac amyloid. Recurrent non-sustained ventricular tachycardia (**panel** (**A**)) is associated with moderate LV symmetric hypertrophy at CMR (**panel** (**B**), MWT 17 mm), thickening of right ventricular free wall and “diffuse“ LGE pattern (**panel** (**C**)). Apple-green birefringent material at polarized light of Congo Red-stained myocardial sections is visible in insert in **panel** (**B**). Moderate infiltration of CT (arrows, **panel** (**D**) hematoxylin and eosin, (**E**) Congo Red staining, (**F**) immunohistochemistry for HCN4, 400× magnification). CT = conduction tissue.

**Figure 4 jcm-12-01798-f004:**
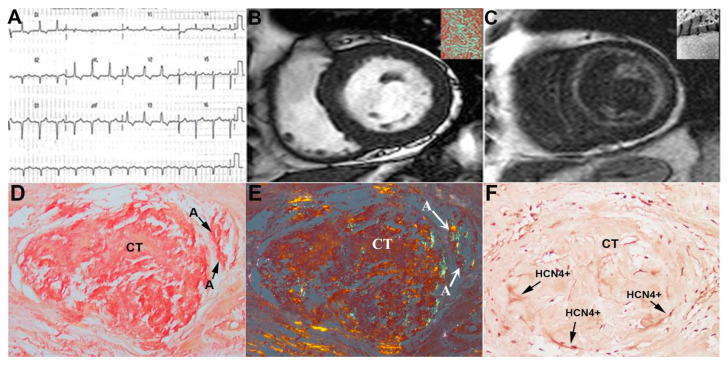
Severe infiltration of CT in cardiac amyloid. Cardiac amyloid presenting with remarkable conduction abnormalities at ECG (right bundle branch block + left anterior fascicular block +I degree A-V block (**panel** (**A**)) and mildly thickened LV (**panel** (**B**), MWT 13 mm) with diffuse “zebroid” LGE at CMR (**panel** (**C**)). Histology with Congo Red staining (**panel** (**D**,**E**) without and with polarized light) and immunohistochemistry for HCN4 (**panel** (**F**)) show massive infiltration of CT involving CT artery (a). Apple-green birefringent material at polarized light of Congo Red-stained myocardial sections is visible in insert in **panel** (**B**). Inserts in **panel** (**C**) suggest ultrastructural evidence of amyloidosis of the myocardium. A = arteriole CT = conduction tissue Black Arrows indicate the positivity immunohistochemistry for HCN4.

**Table 1 jcm-12-01798-t001:** Clinical data and CT infiltration in 17 patients with cardiac amyloidosis.

	Age/Sex	Amyloid Type	2D-Echocardiogram Parameters			Conduction Tissue Infiltration Grading/%	Tachy- Arrhythmias		Brady- Arrhythmias	Therapy
			LVMWT (mm)	LVEF (%)	Diastolic Dysfunction		V (Lown Class)	SV		
1	71/F	AL	17	60	Type II	Mild/22%	1			CHT
2	58/F	AL	17	58	Type II	Mild/20%	1			CHT
3	75/F	AL	17	56	Type II	Severe/80%	3			CHT
4	69/M	AL	15	46	Type I	Severe/85%	4A		3° AV block	PM + CHT
5	57/M	AL	15	60	Type I	Severe/75%	3			CHT
6	70/F	TTR	18	64	Type II	Mild/15%	1			
7	58/F	AL	16	55	Type II	Moderate/65%	4A			Amiodarone + CHT
8	61/F	AL	21	50	Type II	Severe/90%	4A	AF		DC shock + Amiodarone + CHT
9	74/M	AL	16	45	Type II	Moderate/40%	2			CHT
10	81/M	TTR	22	35	Type III	Severe/90%	4A	AF		ICD + amiodarone
11	54/F	AL	15	60	Type III	Mild/25%	1			CHT
12	52/F	AL	12	50	Type I	Severe/85%	3		3-fascicular block	PM + CHT
13	60/M	AL	17	55	Type I	Moderate/55%	2			CHT
14	68/M	TTR	16	50	Type III	Severe/80%	4B	AT	Complete AV block	PM + amiodarone
15	48/F	AL	17	25	Type III	Mild/20%	1			ICD + CHT
16	63/M	TTR	21	50	Type III	Severe/75%	3	Atrial flutter		DC shock + flecainide + bisoprolol
17	72/M	AL	17	55	Type II	Severe/80%	4B	Paroxystic AF		DC shock + amiodarone + CHT

LVMWT = maximum wall thickness; LVEF = ejection fraction; V = ventricular arrhythmias; SV = supraventricular arrhythmias; AT = atrial tachycardia; AF = atrial fibrillation; TTR = transthyretin; AL = amyloid light chain; CHT = chemotherapy; DC = direct current; ICD = Implantable Cardioverter Defibrillator; PM = pacemaker.

## Data Availability

The datasets used and analyzed during the current study are available from the corresponding author upon reasonable request.

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
