# Peer review of "Infiltration of Conduction Tissue Is a Major Cause of Electrical Instability in Cardiac Amyloidosis"

_jcm, 2023, doi:10.3390/jcm12051798_

Round 1

Reviewer 1 Report

The article "Pathology of Conduction Tissue in Cardiac Amyloid:1 Correlation with Arrhythmic Manifestations" is about an interesting, and clinically relevant topic. The study and its goals can be followed easily as described by the authors.

However, this reviewer has the following recommendations about the manuscript:

The manuscript should be checked by a native English speaker, because in several paragraphs the English is not very clear (although it can be understood) scientifically, and it takes much away from the value, and enjoyability of the manuscript.

This reviewer has not pointed out the typos and grammatical errors through the manuscript. This is not the job of a reviewer.

At 126th line, there is an unnecessary line break.

Figure 1 E and Figure 3 E: This reviewer advises the use of white labels and arrows, because of the dark background in the photos.

Figure 1 and Figure 3: please, use of a scale bar, it cannot be seen on the tissue photos.

This reviewer would avoid using angstrom (Å) in the manuscript since it's a former (out of date) unit. This reviewer would use nanometer (nm).

Author Response

The article "Pathology of Conduction Tissue in Cardiac Amyloid:

Correlation with Arrhythmic Manifestations" is about an interesting, and clinically relevant topic. The study and its goals can be followed easily as described by the authors.

However, this reviewer has the following recommendations about the manuscript:

 The manuscript should be checked by a native English speaker, because in several paragraphs the English is not very clear (although it can be understood) scientifically, and it takes much away from the value, and enjoyability of the manuscript.

 This reviewer has not pointed out the typos and grammatical errors through the manuscript. This is not the job of a reviewer.

At 126th line, there is an unnecessary line break.

Reply: the corrections have been made

Figure 1 E and Figure 3 E: This reviewer advises the use of white labels and arrows, because of the dark background in the photos.

Reply: As suggested we added white arrows and labels

Figure 1 and Figure 3: please, use of a scale bar, it cannot be seen on the tissue photos.

Reply:

 This reviewer would avoid using angstrom (Å) in the manuscript since it's a former (out of date) unit. This reviewer would use nanometer (nm).

Reply: As suggested the corrections have been made

Reviewer 2 Report

1. The "Title" could be a bit more "catchy", triggering, and the "Short Title" could be more concise.

2. In Lines 74-76, this reads more like a conclusion than a purpose.

3. How to define "a clinical phenotype of restrictive cardiomyopathy", explain it.

4. Why are the remaining 28 individuals excluded from the present study, because histological sections of conduction tissue are not available? What are the indications for biopsy of heart tissue? Study flowchart is suggested.

5. Is this invasive test routinely performed in patients with restricted cardiac disease? Or are you doing it for research purposes?

6. For Table 1, it is suggested to combine the Conduction tissue infiltration into one column and reduce some unnecessary symbolic annotations for easier reading.

7. The clinical result is suggested if possible.

8. The paper would benefit from some rewording, in particular the conclusion. These evidence tend to show clinical phenomena but can not obtain the conclusion of an independent correlation between them

9. There are a few formatting errors that should be noted.

Author Response

  1. The "Title" could be a bit more "catchy", triggering, and the "Short Title" could be more concise.

Reply:  Report title has been changed into ‘’ Infiltration of Conduction Tissue is a Major Cause of Electrical Instability in Cardiac Amyloidosis’’.

With the following Short title: ‘’ Arrhythmias in Cardiac Amyloidosis’’

  1. In Lines 74-76, this reads more like a conclusion than a purpose. “CT involvement is correlated with type and severity of CA and then with the arrhythmic manifestations”.

Reply: the sentence has been removed

  1. How to define "a clinical phenotype of restrictive cardiomyopathy", explain it.

Reply: The clinical phenotype of restrictive cardiomyopathy has been defined according to the ESC definition; the relative reference has been added. Rapezzi C, Aimo A, Barison A, Emdin M, Porcari A, Linhart A, Keren A, Merlo M, Sinagra G. Eur Heart J. 2022. Restrictive cardiomyopathy: definition and diagnosis. Dec 1;43(45):4679-4693. doi: 10.1093/eurheartj/ehac543.

  1. Why are the remaining 28 individuals excluded from the present study, because histological sections of conduction tissue are not available? What are the indications for biopsy of heart tissue? Study flowchart is suggested.

Reply: The remaining 28 individuals with cardiac amyloid were excluded from the present study, as their biopsies failed to contain sections of CT. Study flowchart has been added.

  1. Is this invasive test routinely performed in patients with restricted cardiac disease? Or are you doing it for research purposes?

Reply: Our is a tertiary referring centre for the diagnosis and treatment of Cardiomyopathies , myocarditis and heart failure. Endomyocardial biopsy represents a common investigational approach for histological and molecular characterization of myocardial tissue. Since 1982, I have personally performed >7000 procedures with 0% mortality and transient complications <1% (1). EMB is considered a safe procedure if obtained by experts working in a hospital provided by cardiac surgery support. I have contributed to a scientific statement on use of endomyocardial biopsy that has been approved concurrently by American College of Cardiology, American Heart Association and European Society of Cardiology and published in 2007 on JACC, Circulation and EUR Heart J (1,2,3).

  1. Leslie T Cooper, Kenneth L Baughman, Arthur M Feldman, Andrea Frustaci, Mariell Jessup, Uwe Kuhl, Glenn N Levine, Jagat Narula, Randall C Starling, Jeffrey Towbin, Renu Virmani; American Heart Association; American College of Cardiology; European Society of Cardiology; Heart Failure Society of America; Heart Failure Association of the European Society of Cardiology. The role of endomyocardial biopsy in the management of cardiovascular disease: a scientific statement from the American Heart Association, the American College of Cardiology, and the European Society of Cardiology. Endorsed by the Heart Failure Society of America and the Heart Failure Association of the European Society of Cardiology. J Am Coll Cardiol. 2007 Nov 6;50(19):1914-31. doi: 10.1016/j.jacc.2007.09.008.
  2. Leslie T Cooper , Kenneth L Baughman, Arthur M Feldman, Andrea Frustaci, Mariell Jessup, Uwe Kuhl, Glenn N Levine, Jagat Narula, Randall C Starling, Jeffrey Towbin, Renu Virmani; American Heart Association; American College of Cardiology; European Society of Cardiology. The role of endomyocardial biopsy in the management of cardiovascular disease: a scientific statement from the American Heart Association, the American College of Cardiology, and the European Society of Cardiology. Circulation. 2007 Nov 6;116(19):2216-33. doi: 10.1161/CIRCULATIONAHA.107.186093.
  3. Cooper LT, Baughman KL, Feldman AM, Frustaci A, Jessup M, Kuhl U, Levine GN, Narula J, Starling RC, Towbin J, Virmani R. The role of endomyocardial biopsy in the management of cardiovascular disease: a scientific statement from the American Heart Association, the American College of Cardiology, and the European Society of Cardiology Endorsed by the Heart Failure Society of America and the Heart Failure Association of the European Society of Cardiology. Eur Heart J. 2007 Dec;28(24):3076-93. doi: 10.1093/eurheartj/ehm456.
  4. For Table 1, it is suggested to combine the “Conduction tissue infiltration” into one column and reduce some unnecessary symbolic annotations for easier reading.

Reply: The Table has been modifies as suggested.

  1. The clinical result is suggested if possible.

Reply: it has been added in the Discussion “Anyway, our study demonstrates that the occurrence of major brady or tachy-arrhythmias in patients with cardiac amyloidosis reflects a severe CT infiltration and requires an appropriate investigation even with an electrophysiological study in order to provide a prompt correction with anti-arrhythimic drugs and/or PM/ICD implantation”.

  1. The paper would benefit from some rewording, in particular the conclusion. These evidence tend to show clinical phenomena but can not obtain the conclusion of an independent correlation between them

Reply: the conclusion has been modified.

  1. There are a few formatting errors that should be noted.

Reply: the corrections have been made.

Reviewer 3 Report

Dear authors,

I have received for review your proposed article entitled Pathology of Conduction Tissue in Cardiac Amyloid: Correlation with Arrhythmic Manifestations

I would like to appreciate the effort made by the authors' collective throughout the study and in the writing of this manuscript. Cardiac amyloidosis has become an increasingly studied field in recent years, with important progress in treatment, especially for the ATTR form. The manuscript proposed for review brings to light an understudied correlation that could provide new prognostic clues and therapeutic recommendations for patients with CA. I would like to appreciate the quality of the histological images and the perseverance in enrolling patients over such a long period of time.

However, I must point out a few aspects that need to be corrected, both technical and in terms of content.

Regarding the editing:

·       Lines 126-127 - the text is out of order.

·       In Table 1, I suggest replacing the symbols in the legend with the abbreviation and its meaning. I find the use of symbols as too complicated and counterintuitive in reading the table.

·       Regarding Figure 3, Panel A is almost illegible. Better image quality would help. Also, the abbreviations used should be included as a legend below the figure or in the text (e.g. BBDx - note valid for the whole article).

·       Rows 216, 221, at the end of the first sentence there is an additional space inserted.

·       Rows 267, 281, 282, 284, 317, an extra space is inserted in the sentence.

·       Line 304, an extra space is inserted before the comma.

Regarding the content of the material:

1. Concerning the mentioned patient with severely diminished left ventricular systolic function. What is the explanation for the 25% fraction?

2. Are the patients' survival data known?

3. Regarding the occurrence of arrhythmias: how far after the endomyocardial biopsy did the rhythm/ conduction disturbances occur? Can we exclude an iatrogenically induced disorder as a result of the biopsy?

4. Can the used therapeutic agents be specified? Do you consider that the treatment had a protective effect in patients who did not develop rhythm/ conduction disorders? All the more so as significant progress has been made in recent years especially in the case of ATTR form.

 5. Concerning the discussion part - regarding the statement in line 305: "Likewise no correlation was observed between degree of CT infiltration and type of amyloidogenic protein . It seems that single mutated protein might present specific physic-chemical affinity for molecular components of CT promoting an elective amyloid deposition." can you further elaborate and issue a hypothesis for which no difference between cardiac amyloidosis types has been observed? Also, I appreciate the discussion as too brief, with the recommendation to elaborate on the clinical implications, relationship to treatment and the importance of the study in identifying CA patients with increased risk to develop CT dysfunction.

6. I believe that the Clinical Implications section does not provide an idea of a new approach. I recommend rewriting it or dropping this section completely.

7. Regarding the conclusions, I appreciate that the phrase "Life threatening cardiac arrhythmias may occur along all the progressive phases of human CA." is too general and does not highlight the results of the study. It should be rewritten in such a way as to emphasize that although we are discussing a general risk for these patients, depending on the level of CT infiltration the risk will be higher - ideally a future study direction should be introduced, which would allow the assessment of this infiltration through clinical/paraclinical data.

8. Given the size of the study group, long enrolment time with different therapeutic options between patients enrolled at the beginning of the study compared to those enrolled recently, I consider that a section on study limitations should be included in the manuscript.

9. Although the subject concerns a rare disease, I consider that the bibliographical references could be improved by increasing their number.

Finally, I would like to stress that I appreciate the proposed article as a valuable one, and I look forward to its revised version.

Good luck in all!

Author Response

Dear authors,

I have received for review your proposed article entitled Pathology of Conduction Tissue in Cardiac Amyloid: Correlation with Arrhythmic Manifestations

I would like to appreciate the effort made by the authors' collective throughout the study and in the writing of this manuscript. Cardiac amyloidosis has become an increasingly studied field in recent years, with important progress in treatment, especially for the ATTR form. The manuscript proposed for review brings to light an understudied correlation that could provide new prognostic clues and therapeutic recommendations for patients with CA. I would like to appreciate the quality of the histological images and the perseverance in enrolling patients over such a long period of time.

However, I must point out a few aspects that need to be corrected, both technical and in terms of content.

 Regarding the editing:

  • Lines 126-127 - the text is out of order.

Reply: The corrections have been made

  • In Table 1, I suggest replacing the symbols in the legend with the abbreviation and its meaning. I find the use of symbols as too complicated and counterintuitive in reading the table.

Reply: The Table has been modified as suggested

  • Regarding Figure 3, Panel A is almost illegible. Better image quality would help. Also, the abbreviations used should be included as a legend below the figure or in the text (e.g. BBDx - note valid for the whole article).

Reply: The quality of the figure has been improved and the corrections have been made

  • Rows 216, 221, at the end of the first sentence there is an additional space inserted.

Reply: the corrections have been made

  • Rows 267, 281, 282, 284, 317, an extra space is inserted in the sentence.

Reply: the corrections have been made

  • Line 304, an extra space is inserted before the comma.

Reply: the corrections have been made

 Regarding the content of the material:

  1. Concerning the mentioned patient with severely diminished left ventricular systolic function. What is the explanation for the 25% fraction?

Reply: This patient showed at histology a remarkable infiltration by amyloid of intramural coronary vessels with severe lumen narrowing suggesting an overlapping ischemic damage.

  1. Are the patients' survival data known?

Reply: No details.

  1. Regarding the occurrence of arrhythmias: how far after the endomyocardial biopsy did the rhythm/ conduction disturbances occur? Can we exclude an iatrogenically induced disorder as a result of the biopsy?

Reply: Cardiac arrhythmias were documented at ECG/Holter before execution of EMB. We did not report new arrhythmias during or after this procedure.

  1. Can the used therapeutic agents be specified? Do you consider that the treatment had a protective effect in patients who did not develop rhythm/ conduction disorders? All the more so as significant progress has been made in recent years especially in the case of ATTR form.

Reply: Unfortunately we have not follow-up of the clinical data to answer the question.

  1. Concerning the discussion part - regarding the statement in line 305: "Likewise no correlation was observed between degree of CT infiltration and type of amyloidogenic protein. It seems that single mutated protein might present specific physic-chemical affinity for molecular components of CT promoting an elective amyloid deposition." can you further elaborate and issue a hypothesis for which no difference between cardiac amyloidosis types has been observed? Also, I appreciate the discussion as too brief, with the recommendation to elaborate on the clinical implications, relationship to treatment and the importance of the study in identifying CA patients with increased risk to develop CT dysfunction.

Reply: In our study we failed to report a correlation between maximal left ventricular wall thickening, low QRS voltages at ECG, severity of amyloid infiltration at histology and extent of CT involvement . This may suggest a different affinity of amyloidogenic protein for CT that is provided of a specific molecular pattern as HCN4. We tried to enrich the discussion. It has been underlined that no correlation could be observed between AL and TTR amyloid in terms of severity of CT infiltration (page 9 Line 215-216).

  1. I believe that the Clinical Implications section does not provide an idea of a new approach. I recommend rewriting it or dropping this section completely.

Reply: This section has been removed and replaced by ‘’Limitations of the study’’.

  1. Regarding the conclusions, I appreciate that the phrase "Life threatening cardiac arrhythmias may occur along all the progressive phases of human CA." is too general and does not highlight the results of the study. It should be rewritten in such a way as to emphasize that although we are discussing a general risk for these patients, depending on the level of CT infiltration the risk will be higher - ideally a future study direction should be introduced, which would allow the assessment of this infiltration through clinical/paraclinical data.

Reply: Significance of the occurrence of electrical instability in patients with CA, their relations with CT infiltration and the opportunity for a prompt electrical correction with PM/ICD implantation, is reported in the Discussion.

  1. Given the size of the study group, long enrolment time with different therapeutic options between patients enrolled at the beginning of the study compared to those enrolled recently, I consider that a section on study limitations should be included in the manuscript.

Reply: A new section ‘’Limitations of the study’, has now been introduced in the text.

  1. Although the subject concerns a rare disease, I consider that the bibliographical references could be improved by increasing their number.

Reply: Bibliographical references have been implemented.

 Finally, I would like to stress that I appreciate the proposed article as a valuable one, and I look forward to its revised version.

Reply: We very much appreciate the final reviewer’s considerations.

Round 2

Reviewer 3 Report

Dear authors,

I think that the submitted manuscript has now received the needed corrections.

Congratulations for all your effort and I am looking forward to your future works. 

Kind regards